# Carbon Storage in Old-Growth Homestead Windbreaks of Small Islands in Okinawa: Toward the Sustainable Management and Conservation

**Bixia Chen** [1,2,*] **and Yi-Chung Wang** [3]

1    Faculty of Agriculture, University of the Ryukyus, 1 Senbaru, Nishihara Town, Okinawa Prefecture 903-0213, Japan
2    The United Graduate School of Agricultural Sciences, Kagoshima University, 1-21-24, Korimoto, Kagoshima 890-0065, Japan
3    Department of Forestry and Nature Conservation, Chinese Culture University, 55 Hwa-Kang Road, Shilin District, Taipei City 11192, Taiwan; ycwang1476@gmail.com
*    Correspondence: chenbx@agr.u-ryukyu.ac.jp; Tel.: +81-98-895-8773

**Abstract:** Research Highlights: This study contributes to the improvement of the understanding of ecosystem functions of trees growing outside the forest, by quantifying the carbon sequestration function of a homestead windbreak, for example, a linear forest belt planted bordering a farmhouse in small islands. Background and objectives: Carbon storage in small-scale stands of forests have been less studied compared to that in large-scale forests. The aims of the present study were to clarify the ecological functions of carbon storage and the economic value of homestead windbreaks to propose effective conservation strategies for old-growth homestead windbreaks in the face of climate change. Materials and Methods: On the small islands of Okinawa Prefecture, the dominant tree species used for the homestead windbreaks is fukugi (*Garcinia subelliptica* Merr.). We collected data on the diameter at breast height (DBH) and the height of 23,518 fukugi trees in 10 villages from 2009 to 2018. Results: The total amount of carbon stored in the remnant fukugi homestead trees of the 10 surveyed hamlets was 6089 t-$CO_2$. The amount is equivalent to the carbon amount that is stored in a 40-year-old Japanese cedar (*Cryptomeria japonica*) forest, a representative tree species in Japan, of 20.9 ha area. Furthermore, the estimated economic value of the homestead trees was equivalent to USD 235,433, in terms of the plantation and management costs of 40-year-old Japanese cedar forests. This study revealed that homestead trees planted in an orderly line usually have a high density; hence, they have a high potential for biomass accumulation, carbon sequestration, and climate change mitigation. Moreover, homestead trees could contribute to a reduction in carbon diffusion, by cooling the house and reducing potential energy consumption. The findings related to homestead trees are consistent with those of other types of trees outside forests or small patches of trees: not adding to future land use competition and highly effective at carbon sequestration. Conclusions: The finding related to the carbon storage of homestead trees will provide basic information, as well as a new perspective on future local conservation and its contribution to climate change mitigation. This study suggests the necessity of the existing trees being properly managed, recruiting trees to be planted to replace old-growth trees, and replanting trees near newly established houses or old homesteads where trees have been cut.

**Keywords:** climate change; ecosystem services; hedgerow; small-scale forest management; trees outside the forest

## 1. Introduction

According to the Millennium Ecosystem Assessment [1], ecosystem services provided by forests include climate change mitigation and adaptation, carbon sequestration, hydrological services, support for agricultural productivity, reduced erosion, and increased wildlife habitat and forest products. International climate change agreements allow the carbon stored by afforestation and reforestation to be used to offset $CO_2$ emissions under the Kyoto Protocol [2]. Carbon sequestration refers to the capture and secure storage of carbon that would otherwise be emitted or remain in the atmosphere.

Forests have recently gained popularity as a climate change adaptation/mitigation measure. In addition to direct carbon storage and sequestration, urban trees can contribute to offsetting carbon emission in urban areas. Planting trees in energy-conserving locations around buildings [3] can reduce building energy use and, consequently, emissions from power plants. Estimating forest carbon storage can provide managers and local policymakers with an indicator to assess sustainable management objectives and achieve the target of conservation, as public awareness of global warming may help to increase the importance of forests and forestry-related activities [2,4].

Carbon storage of large-scale forests at both the global [5,6] and national [7–9] levels has gained intensive attention and is relatively well documented. In contrast, carbon storage in small-scale stands of forests has limited studies, except in few cases, e.g., forest patches in agricultural landscapes [10], agroforestry [11–13], and home gardens [14].

The trees outside the forest include the trees on land not classified as forest or other wooded lands, on farms, along roads, and in many other locations, which are not defined as forests in the country-level statistics or by the Food and Agriculture Organization of the United Nations (FAO) [15]. The trees outside the forest are often poorly defined by managers and are mostly absent from official statistics regarding forests and development policies; thus, they need to be better assessed and valued [16].

Homestead trees on the Okinawa Islands, which are composed of trees planted at the borderline/periphery of a private mansion and primarily function as a windbreak [16], are private properties and are by definition not a forest in Japan. On the small islands of Okinawa Prefecture, the dominant tree species used for the homestead windbreaks is fukugi (*Garcinia subelliptica*). Okinawa suffers from numerous typhoons in the summer and strong monsoonal winds in the winter; therefore, these trees were planted on all sides of the homestead to protect the traditional timber house from strong winds. Selective cutting was undertaken to extract timber in the past, when timber was in short supply. There are many remnant huge and old-growth trees prevalent on the Okinawa Islands [16]. The homestead trees make up only a small portion (area) of urban forest in comparison with the continuous or large-scale urban forests; however, they present a harmonized co-existence of people and ecosystem, in terms of their significant functions of carbon storage, provision of wildlife habitat, improving air and water quality, reducing energy consumption, providing recreational opportunities, and enhancing community well-being. These homestead trees qualify as small nature features, both biotic and abiotic, which provide a substantial contribution to ecological processes or biodiversity disproportionately relative to their small size [17].

Huge/old trees usually provide a significant ecological service that cannot be fulfilled by young trees [18,19], e.g., carbon storage [20]. A study showed that huge/old trees were significant contributors to the total amount of carbon stored; however, they are sensitive to climate change, which affects the tree biomass storage [21]. However, homestead trees have been jeopardized by natural hazards, including typhoons and termites, and human factors, such as clear-cutting [22,23].

Thus, the purpose of the present study was to clarify the ecological functions of carbon storage and economic value of homestead windbreaks in terms of carbon storage function to propose effective conservation strategies for old-growth homestead windbreaks in the face of climate change. Specifically, (1) we used tree measurement data to estimate the total biomass and total carbon stored in the homestead fukugi trees in the Okinawa Islands, (2) compared the carbon storage of homestead windbreaks with Japanese cedar (*Cryptomeria japonica*), and (3) calculated the economic value in terms of carbon storage function of these homestead windbreaks, by comparing them with the plantation and management

costs of a planted Japanese cedar forest. Japanese cedar was used for the comparison of carbon storage function and economic value, as there is a lack of related data for fukugi (*Garcinia subelliptica*) trees. The findings of this study can inform local policymakers with quantitative data of the ecological and economic values of homestead windbreaks. Public awareness regarding global warming is growing rapidly, and our findings will help promote the appreciation of greening landscapes by the public.

## 2. Materials and Methods

### 2.1. Survey Sites

Okinawa Prefecture, located at the southernmost part of Japan, encompasses two-thirds of the Ryukyu Archipelago, extending over 1000 km long. The inhabited islands are typically divided into three island groups, Okinawa Islands, Miyako Islands, and Yaeyama Islands. The Miyako Islands and Yaeyama Islands are also called Sakishima Islands. Okinawa Prefecture has an area of 2280.9 km$^2$. The population was 14.5 million in July 2019.

The climate of Okinawa is influenced by the latitude, surrounding ocean, the Black Current, monsoon, and typhoons. In the long term, the annual average temperature of the Okinawa area has been rising at a rate of 1.16 °C per 100 years [24]. The extreme high temperature has increased, while the extreme low temperature has decreased in the past 100 years [24].

The sites surveyed in the present study were distributed across the archipelago. We selected 10 sites (Figure 1 and Table 1), which have the best-preserved homestead trees, for data collection and analysis. Five sites are located on the Okinawa Islands toward the north of Okinawa Prefecture, and another five sites are on the Sakishima Islands, the southernmost part of Okinawa Prefecture. The three islands of Tonaki, Aguni, and Tarama are among the five least developed islands in the Okinawa Prefecture. Taketomi Island is situated near Ishigaki Island, which is the third biggest island in the Okinawa Prefecture and has become a popular tourist destination. Bise hamlet has been of interest to tourists recently, and the tourism industry has grown rapidly [25].

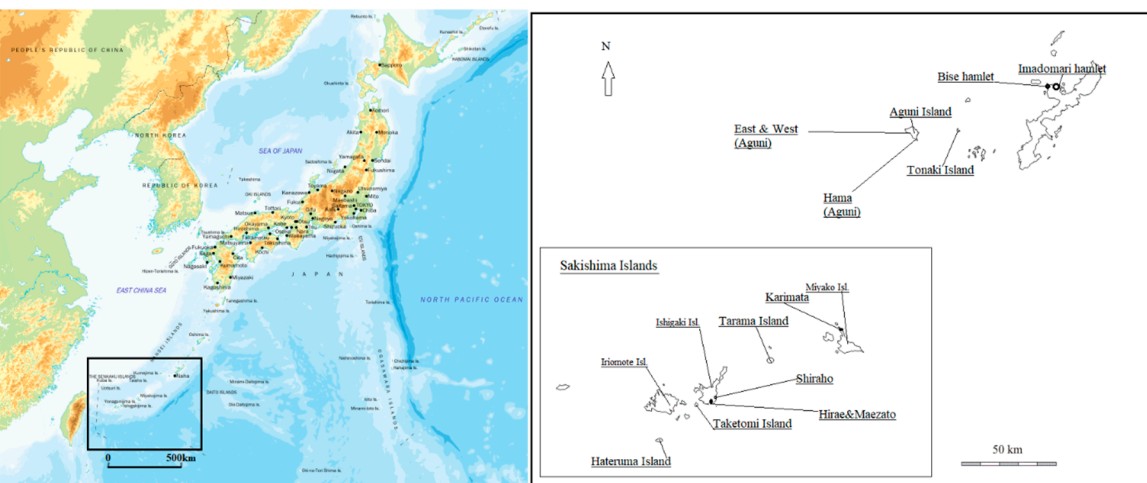

**Figure 1.** Locations of the survey sites.

**Table 1.** Meteorological, natural, and demographic descriptions of survey sites.

| Survey Sites | | | GPS Location | Average Yearly Rainfall | Average Temperature | Wind Velocity | | | Soil Type | Population |
|---|---|---|---|---|---|---|---|---|---|---|
| | | | | | | Average Wind Speed | Days of Wind Speed ≥ 10.0 m/s | Most Frequent Wind Direction | | |
| Okinawa Islands | Kunigami District, Okinawa Island | Bise Hamlet, Motobu Town | N 26°42′ E 127°52′ | 2018.9 (1981–2010) [a] | 22.6 [a] | 3.7 [a] | 24.9 [a] | North-northeast [c] | Shimajiri Mahji [c] | 502 (Nov. 2018) |
| | | Imadomari Hamlet, Nakijin Village | N 26°42′ E 127°55′ | 2108.9 (1981–2010) [a] | 22.6 [a] | 3.7 [a] | 24.9 [a] | North-northeast [c] | Shimajiri Mahji | 891 (Apr. 2018) |
| | Shimajiri District | Tonaki Island | N 26°21′ E 127°08′ | 1860.5 (2015–2018) | | | | North-northeast | Shimajiri Mahji | 378 (Feb. 2018) |
| | | Aguni Island | N 26°35′ E 127°13′ | 1846.4 (2003–2010) | 23.1 (2003–2010) | 4.4 | 25.8 | North | Shimajiri Mahji | 707 (Feb. 2018) |
| Yaeyama Islands | Miyako City | Karimata | N 24°39′ E 125°16′ | 2021 | 23.8 (1981–2010) | 4.7 | 37.3 | Northeast | Shimajiri Mahji | 592 (Dec. 2016) |
| | | Tarama Island | N 24°39′ E 124°14′ | 1986.8(2003–2010) | 24.1 | 4.7 | 58 | North-northeast | Shimajiri Mahji | 1163 (Aug. 2018) |
| | Ishigaki City | Shiraho | N 24°21′ E 124°42′ | 2106.8 (1981–2010) | 24.3 (1981–2010) | 5.5 | 45.2 | North-northeast | Shimajiri Mahji | 816 (Dec. 2016) |
| | | Hirae&Maezato | N 24°53′ E 124°10′ | 2106.8 (1981–2010) | 24.3 (1981–2010) | 5.5 | 45.2 | North-northeast | Shimajiri Mahji | 3769 (Dec. 2016) |
| | Taketomi Town | Taketomi Island | N 24°19′ E 124°05′ | 2106.8 (1981–2010) [b] | 24.3 | 5.5 | 45.2 | North-northeast | Shimajiri Mahji | 305 (Aug. 2018) |
| | | Hateruma Island | N 24°2′ E 123°47′ | 1789.7 (1981–2010) | 24.1 | 4.8 | 42.8 | Northeast | Shimajiri Mahji | 496 (Mar. 2018) |

Note: Meteorological data were sourced from the Japan Meteorological Agency (https://www.jma.go.jp/jma/). Soil type data were sourced from Okinawa General Bureau, Cabinet Office, Japan (http://www.ogb.go.jp/). [a] Data observed in Nago City, approximately 15 km from Bise and 12 km from Imadomari. [b] Data source: from Maezato, approximately 10 km from Taketomi. [c] Shimajiri Mahji: one of the four major soil types in Okinawa. It is an alkalescent dark-red soil and has high water-holding capacity.

## 2.2. Data Collection

As part of conservation activities, an inventory of all residence trees (with a diameter at breast height (DBH) larger than 5 cm) in the hamlets/town cities (that were once villages and have now become downtown areas due to urban sprawl) on the islands of Okinawa Prefecture has been undertaken [15,25,26]. Knowledge of the size of these populations of large old trees and where the trees occur is critical for guiding management [17,27]. The information related to tree sizes, tree heights, and location in each private residence can be used to monitor how and where old-growth trees might change in the future. Tree distribution maps based on different tree measurement methods have been completed in several hamlets [15,26,28], which increase residents' pride in these old trees.

The present study used field data to estimate the total biomass, carbon storage, and sequestration of the homestead trees. Field surveys were conducted to obtain the following characteristics of each tree: DBH, height, and cardinal direction within each village from 2008 to 2018. However, only the parameters of DBH and tree height have been included in the present study.

The dominant tree species used for the homestead windbreaks is *Garcinia subelliptica* (henceforth, the local name of fukugi is used) [15]. Fukugi is a highly distinctive tree that has only one main trunk from which alternating pairs of erect branches arise, giving it a compact, conical crown. Due to its compact upright form, it is planted as a windbreak in Okinawa [29]. In the present study, all fukugi trees in the villages were surveyed and the DBH was measured at a height of 1.3 m; only trees with a DBH larger than 5 cm were surveyed. DBH was measured with a diameter tape, and tree height was directly measured using a sectional measuring pole. However, a couple of survey sites had missing DBH or tree height data. At the beginning of our project, we only targeted the huge trees with a DBH over 25 cm, and two sites, Bise hamlet and Tonaki hamlet, did not have the tree height data. The missing data of tree height were estimated using a formula presented in Section 2.4. Several sites, including Bise hamlet, Imadomari, Higashi and Nishi, and Hama in Aguni Island did not have the tree data with DBH below 5 cm. University students attended the data collection, with a local volunteer team and the university students assisting in data collection from 2016 to 2017.

## 2.3. Calculation Formula

The biomass of each fukugi tree (minimum tree size = 5 cm DBH) was calculated and the allometric equations published by the local government of Okinawa Prefectural Office [30] were used, based on the guidelines of the Intergovernmental Panel on Climate Change. The present study used the calculation formula as follows:

$$V = BA \times H \times f \qquad (1)$$

where V is the individual tree volume ($m^3$/ha); BA equals the basal area($m^2$/ha); H is the tree height (m); f is the average form factor, and the ratio of the volume of a tree to the product of basal area and height, is 0.45.

The organic carbon of the trees was measured from the biomass of the trees multiplied by the carbon factor. The following equation (2) was used to estimate this:

$$C_{single\ tree} = V \times BD \times BEF \times (1+R) \times CF \times Deduction\ rateCO2_{single\ tree} = C_{single\ tree} \times \frac{44}{12} \qquad (2)$$

where V is the individual tree volume ($m^3$/ha); BD equals the basic wood density (kg dry matter/$m^{-3}$ fresh volume) of 0.56; BEF is the biomass expansion factor, the ratio of aboveground biomass (stem, branches, leaves, and twigs) to the stem biomass, is 1.3; R, root shoot ratio, is the ratio of the root biomass to the aboveground biomass, set to 0.25; CF, the carbon fraction of dry matter, is 0.4691; and the deduction rate is 0.9.

Following the Okinawa Prefectural guideline [30], a deduction rate of 0.9 was applied to the amount sequestered in a single tree and all surveyed trees, because the expected amount stored in the

trees may be overestimated due to weather conditions, natural disasters caused by typhoons, and so on. Hence, a coefficient was added in the Okinawa case study [30].

### 2.4. Estimation of Missing Tree Height Data

The two villages of Bise and Tonaki were surveyed in 2008 and 2009, respectively. Here, only DBH data were collected, as the tree height data were missing. A regression equation (3) [31] was used to predict the absent tree heights in these two villages. Hence, we extracted a parabola regression model developed from the survey data (2385 tree samples) from a survey site in Shiraho Village, Ishigaki Island.

$$H = a + bD + cD^2 \tag{3}$$

where H equals the tree height in meters, rounded to the nearest tenth; D equals the diameter at 1.3 m outside bark in centimeters, rounded to the nearest tenth; and *a, b, c* are the regression coefficients. The software Stata 15 was used to calculate the coefficients and R-square values.

### 2.5. Carbon Sequestration in Fukugi Trees Compared to the Planted Forests of Japanese Cedar

To clarify the significance of fukugi tree function in the carbon storage, we compared the fukugi trees to the planted cedar (*Cryptomeria japonica*), the presentative tree species of Japan. Japanese cedar was selected as a comparison for two reasons. First, Japanese cedar is a major planted tree species, accounting for 44% of the planted forest in Japan [31]. Second, an estimation approach of the carbon storage amount for Japanese cedar has been developed and published on the homepage of the Ministry of Agriculture, Forestry and Fisheries, Japan [32]. A 40-year-old Japanese cedar forest under good management can store 79 tons of carbon per hectare, approximately 290 tons of $CO_2$, assuming there are 1000 stand trees on a hectare [33].

Because the fukugi tree is commonly used as homestead windbreak in Okinawa, it did not have an established market price. Hence, we propose that the plantation and management costs of Japanese cedar, [34] can be used to estimate the economic value of fukugi trees in terms of its function of carbon sequestration.

## 3. Results

The results are presented in the following two groups: the Okinawa Islands, where the capital city of Okinawa Prefecture is located, and the Sakishima Islands, including the Miyako Islands and Yaeyama Islands, which are 287 km and 411 km away from Okinawa Island, respectively.

### 3.1. Height–Diameter Regression Model

The $R^2$ value of the model was 0.487, with a significance of 0.00, indicating that the regression model predicted the dependent variable of tree height significantly well. The $R^2$ value is less than 0.5, although it may be subject to the pruning of homestead trees, which results in a wide range of tree height predictions. The coefficients table is presented in Table 2. The regression equation can be presented as:

$$\text{Tree height} = 2.061 + 0.298 * \text{DBH} - 0.003 * \text{DBH}^2 \tag{4}$$

### 3.2. General Dimensions of Fukugi Trees

A total number of 9406 fukugi trees were surveyed at five survey sites (hamlets) on the Sakishima Islands. The mean DBH ranged from 23 cm to 28 cm and the mean tree height ranged from 6 to 7 m in the different hamlets (Table 3).

**Table 2.** Height–diameter regression model.

| | Model Summary | | | |
|---|---|---|---|---|
| **Model** | **R** | **R Square** | **Adjusted R Square** | **Std. Error of the Estimate** |
| 1 | 0.6979834 | 0.4871808 | 0.4867504 | 1.873669 |

a. Predictors: (Constant), DBH Square, DBH
b. Dependent Variable: tree height

| | Model summery | | | | | | |
|---|---|---|---|---|---|---|---|
| | **Unstandardized Coefficients** | | **Standardized Coefficients** | | | **95% Confidence Interval for B** | |
| | **B** | **Std. Error** | **Beta** | **t** | **Sig.** | **Lower Bound** | **Upper Bound** |
| (Constant) | 2.061 | 0.132 | | 15.625 | 0 | 1.803 | 2.32 |
| DBH | 0.298 | 0.01 | 1.471 | 30.289 | 0 | 0.279 | 0.318 |
| DBH Square | −0.003 | 0 | −0.864 | −17.789 | 0 | −0.003 | −0.003 |

Coefficients (a)

**Table 3.** Carbon storage estimated from tree numbers in Okinawa Islands.

| | Tree Number | Mean DBH (cm) | Mean Tree Height (m) | Mean Individual Tree Volume ($m^3$) | Mean Estimated Carbon Storage in an Individual Fukugi Tree ($t\text{-}CO_2$) | Total Estimated Carbon Storage ($t\text{-}CO_2$) | Equivalent to Japanese Cedar Planted Area [1] (ha) |
|---|---|---|---|---|---|---|---|
| Tonaki.a [2] (DBH ≥ 25cm) | 962 | 31.13 | 8.32 | 0.284959799 | 0.381001012 | 329.870676 | 1.13748509 |
| Tonaki.b [3] (DBH ≥ 5cm) | 7,680 | 17.28 | 6.17 | 0.065114044 | 0.096733008 | 668.6185504 | 2.305581208 |
| Bise [2] | 1,075 | 38.52 | 8.89 | 0.466204377 | 0.692590244 | 670.0810612 | 2.310624349 |
| Imadomari [2] | 1,293 | 35.18 | 9 | 0.393673524 | 0.584838873 | 680.5769965 | 2.346817229 |
| Aguni (Higashi &Nishi) (2) | 2,561 | 33.17 | 7.23 | 0.281145585 | 0.417668086 | 962.6831717 | 3.319597144 |
| Aguni (Hama) [2] | 541 | 31.27 | 7.13 | 0.246403754 | 0.366055844 | 178.2325903 | 0.614595139 |
| Total [4] | 14,112 | | | | | 3160.19237 | 10.89721507 |
| Mean [4] | 2,352 | 24.44658403 | 6.916555894 | 0.179740206 | 0.267020902 | 632.038474 | 2.179443014 |
| S.D. [4] | 2697.9732 | 7.307660136 | 1.125042221 | 0.153580286 | 0.228157892 | 283.0225366 | 0.975939781 |

Note: [1] Refer to the total amount of carbon stored in fukugi tree the survey site equivalent to the $CO_2$ amount stored in a certain area of a 40-year-old Japanese Japanese cedar (Japanese Cedar, *Cryptomeria japonica*) forest. It is assumed that a 40-year-old Japanese cedar forest with 1000 trees in a hectare. [2] Only fukugi trees with DBH larger than 25 cm were measured. [3] Fukugi trees with DBH larger than 5 cm were measured. [4] Tonakia is excluded in these calculations.

A total number of 13,150 fukugi trees were surveyed at five survey sites (hamlets) on the Okinawa Islands. The mean DBH ranged from 17 cm to 38 cm and the mean tree height ranged from 7 to 10 m in the different hamlets. The deviations for DBH and tree height in the Okinawa Islands were much higher than those of the mainland Okinawa, partly due to the difference in survey methods applied. Among the five survey sites, two sites in Bise and Imadomari, and two sites in Aguni Island, only data for trees larger than 25 cm DBH were available. The two sites of Bise and Imadomari are located in the northern part of Okinawa, where we observed fukugi trees that were higher than those on the isolated islands. The wind speeds are much higher at the isolated islands, particularly for the Sakishima Islands, compared to the wind speeds on the Okinawa Islands (Table 1).

### 3.3. Carbon Stock in Fukugi Trees

Carbon sequestered in the fukugi trees in the survey hamlets on the Okinawa Islands was estimated to be 632.0 ($\pm$283.0) t-$CO_2$ on average. Carbon sequestered in the fukugi trees in the survey hamlets on the Sakishima Islands was estimated to be 585 ($\pm$273.7) t-$CO_2$ on average. Therefore, the carbon sequestered in the fukugi trees surveyed on the Okinawa Islands was slightly higher than that on Sakishima Islands on average, because the total biomass was higher in the fukugi trees in the Okinawa Islands than that in the Sakishima Islands. The total carbon amount of the homestead trees in all 10 survey sites was 6089 t-$CO_2$.

The carbon stored in each fukugi tree was 0.27 t-$CO_2$ in the Okinawa Islands and 0.34 t-$CO_2$ in the Sakishima Islands. The difference in average tree size contributed to the difference in carbon storage.

### 3.4. Comparison of Carbon Sequestration between Fukugi Trees and a Japanese Cedar Forest

In terms of the carbon sequestration function, fukugi trees in the surveyed hamlets were equivalent to an area of Japanese cedar forest of 2.2 ($\pm$0.97) ha on the Okinawa Islands. Fukugi trees in the surveyed hamlets on the Sakishima Islands were equivalent to an area of a Japanese cedar forest of 2 ($\pm$0.94) ha. The comparisons are shown in Tables 3 and 4. Fukugi trees in the surveyed homesteads stored a total amount of carbon, equivalent to that of the Japanese cedar forest, accounting for approximately 20.9 ha.

According to the Forestry Management Statistical Survey Report in 2013, it was estimated that the cost of the creation of a 40-year-old Japanese cedar forest plantation, together with its maintenance costs, was approximately JPY 1,194,646 (USD 1 $\approx$ JPY 108) per hectare [35]. Therefore, all the current homestead forests in the total 10 survey sites are worth approximately 24,968,101 JPY, which is equivalent to USD 231,186. Hence, the carbon stored in the fukugi trees could be estimated to be USD 38/ton.

**Table 4.** Total carbon storage data in five survey sites on Yaeyama Islands (Ishigaki Islands and Miyako Islands).

| | Tree Number | Mean DBH (cm) | Mean Tree Height (m) | Mean Individual Tree Volume (m$^3$) | Mean Individual Estimated Carbon Storage (t-CO$_2$) | All Trees (t-CO$_2$) | Deducted Total Estimated Carbon Storage (t-CO$_2$) | Equivalent to Japanese cedar Planted Area [1] (ha) |
|---|---|---|---|---|---|---|---|---|
| Shiraho | 2385 | 24.7 | 7.26 | 0.2394 | 0.3751 | 894.6672094 | 805.2004884 | 2.776553408 |
| Hirae and Maezato | 1700 | 26.57 | 7.28 | 0.2371 | 0.37 | 567.8622932 | 511.0760638 | 1.762331255 |
| Taketomi | 1139 | 23.31 | 6.16 | 0.2177 | 0.3235 | 368.4847603 | 331.6362843 | 1.143573394 |
| Hateruma | 2813 | 28.25 | 7.23 | 0.249 | 0.37 | 1040.886105 | 936.7974948 | 3.230336189 |
| Karimata | 1369 | 23 | 6.44 | 0.188 | 0.2796 | 382.8311 | 344.54799 | 1.188096517 |
| Total | 9406 | - | - | - | - | 3254.731468 | 2929.258321 | 10.10089076 |
| Mean | 1881.2 | 25.68390814 | 7.002093345 | 0.231746577 | 0.339173081 | 650.9462936 | 585.8516643 | 2.020178153 |
| Standard deviation | 701.4465054 | | | | | 304.1488811 | 273.733993 | 0.943910321 |

Note: [1] Refer to the total amount of carbon stored in fukugi tree the survey site equivalent to the CO$_2$ amount stored in a certain area of a 40-year-old Japanese Japanese cedar (Japanese Cedar, *Cryptomeria japonica*) forest.

## 4. Discussion

Our findings add new knowledge to the current global discussions of the role of small-scale forests and trees outside the forests in combating global warming. Consistent with the methods in previous studies on other types of small forests and other areas worldwide [10,12,14,17], we quantified the carbon sequestration in small plots of private forests and the trees outside of the forest, which are neglected and largely lacking in the current international literature. The total amount of organic carbon stored by currently existing fukugi trees in the homesteads was approximately 670 ton inside a hamlet and approximately 6738 t-$CO_2$ in the total ten surveyed sites. A comparison with Japanese cedar forests showed that the total carbon amount stored in the homestead trees at the 10 surveyed sites was equivalent to that of a Japanese cedar forest of 20.9 ha in size.

In this study, homestead trees planted in an orderly linear form usually have a high density to effectively function as windbreak; hence, they have a high potential for biomass accumulation, carbon sequestration, and climate change mitigation, if we can properly manage homestead windbreaks. Proper tree management strategies are needed to maintain the existing trees, plant recruitment trees to replace declining old-growth trees, and replant trees near newly established houses or old homesteads where trees have been removed.

In addition, farm windbreaks can contribute to offset carbon diffusion by potentially reducing fossil-fuel burning by providing shade to houses, thus, decreasing electricity consumption [36,37]. A well-established homestead windbreak can reduce home heating cost by 20–30 percent [38]. Windbreaks provide a promising means to reduce greenhouse gas emissions, by reducing fossil fuel usage from farm operations in US [37]. In particular, energy saving is significant in Okinawa, where approximately 83.7% of the total energy was produced by liquefied natural gas (42.1%), coal (32.3%), and petroleum (9.3%) in Japan in 2016 [39].

The homestead trees are valuable to human well-being and society [36]. They have dramatically changed from providing provisioning services to meet the basic needs of the local people to regulating and culturing services [40]. However, the transformation of the attitudes of residents and their lifestyles, from a rural society to one where the majority of the residents have stopped farming, has resulted in abandoning tree maintenance and cutting [22,25].

From the perspective of the contribution of homestead trees to carbon storage, it has a profound significance, as it will not add pressure to current land use competition. Stopping or slowing deforestation and forest degradation and increased reforestation have been proposed as methods for stabilizing carbon storage and sequestration in the atmosphere [31]. However, these techniques are impractical in the face of an increasing global population and urbanization. Marginal or arid lands, which are not suitable for farming, have a high potential for afforestation [10,13,41,42].

In terms of management and conservation, similar to trees outside of forests, there are challenges and barriers in maintaining homestead trees; however, they are not yet considered an integral part of planning and development policies [15]. Thus, this study provides a potential approach for calculating the contingent economic value of homestead trees, in terms of their ecological role in storing carbon. The value can be then used as an essential parameter of tree conservation planning.

## 5. Conclusions

Looking at old-growth habitat woodlands as a significant carbon storage pool has profound implications for management strategies and conservation goals. The finding related to the carbon storage of homestead trees provides basic information, as well as a new perspective for the future local conservation scheme and connection to the global efforts to combat climate change. Integrating carbon sequestration objectives into village landscape planning or town planning allows managers to assess compliance with sustainability criteria, as well as the opportunity to take advantage of carbon credit trading [4]. However, as the homestead trees belong to many owners, a policy must be designed to treat all the homestead trees in a village as a collective unit and the carbon credit trading benefits to be counted at the village collective, which is also a problem for scattered homestead trees. Generally,

governments have limited authority to regulate private forest management and incentives, e.g., tax credits, subsidies, cost sharing, contracts, technical assistance, and environmental payments [42]. To conserve the climate stabilizing effect of traditional homestead windbreaks and steer necessary conservation measures toward climate-friendly solutions, we suggest that biodiversity and C-rich old-growth homestead trees should be included in current discussions about Reducing Emissions from Deforestation and Forest Degradation (REDD+) and/or their owners should be rewarded for their environmental services through other incentive mechanisms.

For Okinawa, several conservation activities are required. These include: (1) decrease the rates of adult tree mortality and plant new young trees to recruit the old-growth trees. Protection of remnant old-growth trees is the most important action for homestead tree management. The potential recruit trees should also be protected, as they will eventually replace the dead old trees, otherwise there are significant opportunity costs pertinent for reducing tree mortality and increasing tree regeneration [43]. (2) Develop a local strategy for the conservation of homestead windbreaks. A long-term management plan should include local policymakers, community leaders, rural households, and small business owners in integrated windbreak conservation and home garden planning efforts. The conservation strategy should involve local experience and traditional utilization to highlight its practical scope, advantages, and weak points. (3) Model guidelines for the sustainable management of windbreaks should be formulated and integrated into a local ordinance. A homestead windbreak forest management policy is imperative for biodiversity conservation and landscape protection. The economic actors managing the windbreak forests for all inhabitants should be given training and fiscal and financial incentives, as well as appropriate public subsidies, and support for the marketing of products from home gardens should be provided to avoid the clear-cutting of windbreak forests.

**Author Contributions:** Conceptualization, B.C. and Y.-C.W.; methodology, B.C. and Y.-C.W.; software, B.C.; validation, B.C.; formal analysis, B.C.; investigation, B.C.; data curation, B.C. and Y.-C.W.; writing—original draft preparation, B.C.; writing—review and editing, B.C. and Y.-C.W.; visualization, B.C.; supervision, B.C. and Y.-C.W.; project administration, B.C.; funding acquisition, B.C. and Y.-C.W.. All authors have read and agreed to the published version of the manuscript.

**Funding:** This work was partly supported by Research Institute for Islands and Sustainability (RIIS), the University of the Ryukyus and the Japan Society for the Promotion of Science Grant-in-Aid for Scientific Research (C) (No. 16K07781) and (B) (No. 18H01612A).

**Acknowledgments:** The author wants to thank local volunteers led by Chokin Miyara for their assistance with data collection in the field.

**Conflicts of Interest:** The authors declare no conflict of interest.

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
