# Peer review of "Carbon Storage in Old-Growth Homestead Windbreaks of Small Islands in Okinawa: Toward the Sustainable Management and Conservation"

_forests, doi:10.3390/f11040448_

Round 1

Reviewer 1 Report

This is a good manuscript now, congratulations.

Author Response

Journal Forests (ISSN 1999-4907)

Manuscript ID forests-757492

Type Article

Point-by-point responses to the reviewer’s comments on the manuscript

" Carbon storage in old-growth homestead windbreaks of small islands in Okinawa: Toward the sustainable management and conservation"

The authors would like to thank the editor and reviewers for careful review of our manuscript and providing us with their comments and suggestion to improve the quality of the manuscript. The following responses have been prepared to address all the reviewers’ comments in a point –by-point fashion. The revisions to the text are marked in red color.

Response to the comments

Reviewer #1

This is a good manuscript now, congratulations.

Response: We greatly appreciation your kind support to enable the publication of this manuscript

Reviewer 2 Report

The manuscript attempts to quantify the carbon stocks of trees, which are not part of conventional forests and focuses its attention on a native species called Fukugi, however the quantification is done through the better-known Japanese cedar. While this is a bit confusing it is not clear to me why this is not mention in the abstract. Furthermore, the comparison to Japanese cedar carbon stocks, even though it is valid it should be highlighted that the comparison is with Japanese cedar grown in Okinawa, and not for example in Fukushima, Yamagata or Miyagi, where it would have a different growth rate/ age relationship and in consequence a different carbon fixation capabilities.

Abstract

There is no mention of Fukugi, without it, the conclusions are not matched with the abstract.

Line 24: it is better to use Japanese cedar instead of sugi.

Line 25: use only the latin mane.

Line 34-35: …homestead trees provide basic information as well as new perspective on future local conservation and its contribution to climate change mitigation.

Introduction

While reading the introduction, the reader is led to think that the objective of the paper is to quantify the carbon stocks in cedar trees and just in the end, after the objectives there is a mention of the Fukugi trees, which is when I started to feel confused with the flow of the paper. Fukugi tree is an important tree species in urban or village areas and for a better understanding it should be mentioned much earlier in the introduction.

Line 51: … urban trees can contribute to offset emissions in urban areas.

Line 59: … except in few cases….

Line 78: well-being.

Line 91: …with Japanese cedar (Cryptomeria japonica)

Line 92: … by comparing them with…

Line 94-96: This should be brought before the objectives. The reader needs this information in advance.

Line 105: use one decimal. 2280.9, same in Line 109.

Line 125: … where old-groth trees …

Line 164: …due to weather conditions…

Line 175: …to the planted cedar forest.

Line 177 to 181: change all the ‘sugi’ words for ‘cedar’

Line 185: change sugi for cedar.

Line 236: where is Yaeyama Island?

Line 239: Fugi? Do you mean ‘Cedar’?

Line 241: do you know the area occupied by Fukugi trees?

Line 246: do you mean the cost of wood or the cost of carbon exchange?

Discussion

The discussion is a bit short and more could be discussed here in comparison to other windbreaks or homestead trees in other areas.

Line 252: change ‘climate’ for ‘global’

Line 257: change ‘survey’ for ‘surveyed’.

Line 258-259: replace ‘sugi’ for ‘cedar’.

Line 260: do you mean ‘low’ density? And for that reason they have high potential to grow.

Conclusion

The conclusion is too long. You could use part of it in the discussion.

Line 283: …well-being…

Author Response

Journal Forests (ISSN 1999-4907)

Manuscript ID forests-757492

Type Article

Point-by-point responses to the reviewer’s comments on the manuscript

Reviewer #2

The manuscript attempts to quantify the carbon stocks of trees, which are not part of conventional forests and focuses its attention on a native species called Fukugi, however the quantification is done through the better-known Japanese cedar. While this is a bit confusing it is not clear to me why this is not mention in the abstract. Furthermore, the comparison to Japanese cedar carbon stocks, even though it is valid it should be highlighted that the comparison is with Japanese cedar grown in Okinawa, and not for example in Fukushima, Yamagata or Miyagi, where it would have a different growth rate/ age relationship and in consequence a different carbon fixation capabilities.

Response: A description of fukugi has been added in the abstract part as suggested.

we used the data from the cedar trees in Japan, as there is no cedar tree in Okinawa.

Abstract

There is no mention of Fukugi, without it, the conclusions are not matched with the abstract.

Response: Lines 20-21 have been added as suggested.

Line 24: it is better to use Japanese cedar instead of sugi.

Response: Changed as suggested.

Line 25: use only the latin mane.

Response: Changed as suggested.

Line 34-35: …homestead trees provide basic information as well as new perspective on future local conservation and its contribution to climate change mitigation.

Response: Changed as suggested.

Introduction

While reading the introduction, the reader is led to think that the objective of the paper is to quantify the carbon stocks in cedar trees and just in the end, after the objectives there is a mention of the Fukugi trees, which is when I started to feel confused with the flow of the paper. Fukugi tree is an important tree species in urban or village areas and for a better understanding it should be mentioned much earlier in the introduction.

Response: Introduction of fuguki trees has been added

Line 51: … urban trees can contribute to offset emissions in urban areas.

Response: Changed as suggested.

Line 59: … except in few cases….

Response: Changed as suggested.

Line 78: well-being.

Response: Changed as suggested.

Line 91: …with Japanese cedar (Cryptomeria japonica)

Response: Changed as suggested.

Line 92: … by comparing them with…

Response: Changed as suggested.

Line 94-96: This should be brought before the objectives. The reader needs this information in advance.

Response: Changed as suggested.

Line 105: use one decimal. 2280.9, same in Line 109.

Response: Changed as suggested.

Line 125: … where old-groth trees …

Response: Changed as suggested.

Line 164: …due to weather conditions…

Response: Changed as suggested.

Line 175: …to the planted cedar forest.

Response: Changed as suggested.

Line 177 to 181: change all the ‘sugi’ words for ‘cedar’

Response: changed as suggested.

Line 185: change sugi for cedar.

Response: changed as suggested.

Line 236: where is Yaeyama Island?

Response: Changed to Sakishima Islands.

Line 239: Fugi? Do you mean ‘Cedar’?

Response: Changed to Fukugi.

Line 241: do you know the area occupied by Fukugi trees?

Response: We can find out the means to calculate Fukugi trees in the area, because fukugi trees are majorly planted in lines as windbreak.

Line 246: do you mean the cost of wood or the cost of carbon exchange?

 Response: We meant the cost invested for the plantation and maintenance of 40-year-old cedar forests.

Discussion

The discussion is a bit short and more could be discussed here in comparison to other windbreaks or homestead trees in other areas.

Response: the first paragraph in the conclusion part has been moved to the discussion part as suggested. Three literatures pertinent to farm and farmstead windbreaks have been added in Lines 259-262.

Line 252: change ‘climate’ for ‘global’

Response: changed as suggested.

Line 257: change ‘survey’ for ‘surveyed’.

Response: changed as suggested.

Line 258-259: replace ‘sugi’ for ‘cedar’.

 Response: Changed as suggested.

Line 260: do you mean ‘low’ density? And for that reason they have high potential to grow.

Response: Lines 252-254 have been rephrased for the clarification.

Conclusion

The conclusion is too long. You could use part of it in the discussion.

Response: The first paragraph has been moved to the discussion part.

Line 283: …well-being…

Response: Changed as suggested.

Round 2

Reviewer 2 Report

The paper has been greatly improved and i now can recommend it for publication.

There are some minor corrections as listed below'

Line 139: erase 'that'.

Line 165: erase 'in'

Line 202: add ... homeastead fukugi trees...

Line 381-382: Homestead trees are valuable to human well-being and society [36]. They have dramatically changed from....

Author Response

Response to Reviewer 2 Comments

Point 1: The paper has been greatly improved and I now can recommend it for publication.

Response 1: Thanks for the very positive comment to support the publication of our manuscript.

Point 2: There are some minor corrections as listed below'

Response 2: Thanks very much for your corrections which have greatly improved our manuscript quality.

Point 3: Line 139: erase 'that'

Response 3: Line 23-Deleted as suggested

Point 4: Line 165: erase 'in'

Response 4: Line 49- Deleted as suggested.

Point 5: Line 202: add ... homeastead fukugi trees

Response 5: Line 87: Added as suggested.

Point 6: Line 381-382: Homestead trees are valuable to human well-being and society [36]. They have dramatically changed from.

Response 6: Line 265. Changed as suggested.

This manuscript is a resubmission of an earlier submission. The following is a list of the peer review reports and author responses from that submission.

Round 1

Reviewer 1 Report

General comments

The manuscript reports a study on carbon stock and related economic value of trees out of forests and specifically of trees planted as windbreaks in Okinawa. Data were collected over a nine years period, carbon storage was calculated based on common formulae relating tree volume to amount of organic carbon. The analysis carried out in this study is basic. The study seems a report rather than a research paper: reporting basal area calculation and the dbh-height function as a result are an example. Furthermore, there are a number of flaws as the inconsistency in the use of terminology, the presence of entire paragraphs in the wrong sections, and many assumptions not anticipated with a proper explanation (see specific comments). Comparisons between species should be made by comparing values observed for similar ages and climatic conditions. Furthermore, the value of carbon sequestration was based on the cost of management of Japanese cedar and not on carbon market values. Discussion should be expanded by comparing results from the literature on trees outside forest. Finally, English revision is required.

Specific comments

L21-23: I suggest reporting the overall value (maybe a mean per hectare) and then report the equivalent for the Japanese cedar (most common tree plantation? Why this specific species?). Be consistent throughout the text in the use of the species’ common name (sugi? Japanese sugi? Japanese cedar?).

L24-25: Do not understand this comparison? Does the related value of Japanese cedar miss? Or is the added value compared to Japanese cedar?

L58-59: The definition of residence trees is strange. Do they represent “trees outside forest” or do they form “human-made forests”? In this sentence also the use of “residence” should be explained.

L70-77: I suggest moving this part in the method section.

L83-84: It is not clear if authors refer to the total economic value of these trees or only as for carbon storage?

L90: “fukugi trees”?

L102-104: How was this selection carried out? Sakishima Island..but the title specifically refers to Okinawa.

L104-110: Can be deleted. It can be observed from figure 1.

L112-129: Can be shortened.

Formula 1: Please do not report the formula of the basal area!

L143: How was R calculated or derived?

L151-157: Nothing is mentioned on how the model was selected? By the way it seems to be a polynomial functions of the second order and not a linear function!

L163-164: Not clear how this was done?

R-squared is quite low!

Table 4: No per hectare values for the “residence trees” is reported. This would enable comparison with other studies.

Discussion: I would expect comparison with data from other value of trees outside forests. What about comparing different types of trees outside forests?

Reviewer 2 Report

The paper "Carbon storage in old-growth residence windbreaks of small islands in Okinawa: Toward the sustainable management and conservation" deals with an investigation about contribution of the farmstead windbreaks to the carbon budget. In Okinawa Islands, Miyako Islands, and Yaeyama Islands trees were surveyed.

The article topic is interesting.

Considerations on the implications of the studied windbreaks should be added related to sustainable management aspects, also in relation to climate change (as the text suggests).

My major concern is related to the development in the methods of the objectives of this paper. In the aim of the study, the authors declared: to estimate the total biomass and total carbon stored in the residence trees in the Okinawa Islands, to compare the carbon storage of homestead windbreaks with sugi (Japanese Cedar, Cryptomeria japonica), and to calculate the economic value in terms of carbon storage function of these homestead windbreaks. I did not find clear evidence of these in the discussion section. These are an important aspect to be discussed and underlined but the authors did not grant it the necessary space.

Detailed comments.

Title:  The word residence in line 2 must be changed for farmstead and in the whole manuscript

Abstract: Must be improved according to the changes suggested.

Materials and Methods: The author could include the climatologic characteristics of these islands. To develop the methods for each specific objective. To describe the data analysis

Discussion: To include more discussion and at the end of this section make and integration of the all results.

In line 172 the adjusted R squared is very low to make inferences, therefore the author must declare this finding and use some regression remedial measurements to increase this value. Contrary, justify in the discussions.

Conclusions. To summarize it to three, one for each objective.